# A Quantitative Histologic Analysis of Oogenesis in the Flatfish Species *Pleuronectes platessa* as a Tool for Fisheries Management

**DOI:** 10.3390/ani13152506

**Published:** 2023-08-03

**Authors:** Carine Sauger, Jérôme Quinquis, Clothilde Berthelin, Mélanie Lepoittevin, Nicolas Elie, Laurent Dubroca, Kristell Kellner

**Affiliations:** 1Unity Biology of Organisms and Aquatic Ecosystems (UMR 8067 BOREA), University of Caen-Normandie, Muséum National d’Histoire Naturelle, Sorbonne University, CNRS, IRD, Université des Antilles, Esplanade de la Paix, 14032 Caen, France; clothilde.berthelin@unicaen.fr (C.B.); melanie.lepoittevin@unicaen.fr (M.L.); 2Laboratoire Ressources Halieutiques de Port en Bessin, Institut Français de Recherche pour l’Exploitation de la Mer (IFREMER), Avenue du Général de Gaulle, 14520 Port en Bessin Huppain, France; dbklor@gmail.com (J.Q.); laurent.dubroca@ifremer.fr (L.D.); 3Service Unit PLATON, VIRTUAL’HIS, Federative Structure 4207 “Normandie Oncologie”, Normandie University UNICAEN, 14000 Caen, France; nicolas.elie@unicaen.fr

**Keywords:** sexual maturity, histology, fish ovary

## Abstract

**Simple Summary:**

Histology and stereology were used to estimate the sexual maturity phase for the European Plaice, as well as a description of the oogenesis cycle and size at first maturity. This study provides a method to accurately determine a key parameter for fisheries management that may be used for other species.

**Abstract:**

The following paper gives a detailed description of the oogenesis cycle for the European Plaice (*Pleuronectes platessa*), from oogonia to post-ovulatory follicle, including ovarian follicle and *zona pellucida* sizes. Noteworthy particularities were the difficulty in identifying cortical alveoli due to their very small size. Quantitative histology (stereology) on histological slides was used to determine a first size at maturity for females from the English Channel, which was found to be smaller compared to the literature (19 cm). Stereology also determined a first spawning event starting in January, with a peak in February and ongoing until March. Moreover, the use of stereology showed misclassifications for individuals categorized into a maturity phase using a macroscopic visual method. Misclassifications were found with individuals that had spawned (D) but were put under the immature (A) phase, and individuals in development (B) classified under D.

## 1. Introduction

The reproductive capacity, or more precisely, the capacity to produce viable eggs or larvae, is a key parameter for stock management of exploited fish species [1,2]. Subsequently, a fine appreciation of the reproductive biology of targeted species is of great importance when evaluating the productivity of exploited stocks. Regular sampling of these stocks and the determination of parameters linked to reproductive status, such as sex ratio, sexual maturity or fecundity, may contribute to the evaluation of their reproductive capacity. Usually, the reproductive success of a species is mainly attributed to females and is determined by the percentage of mature females (females showing gonad development) in the stock [3]. Estimating this reproductive success is one of the main objectives in fisheries management, with the generation of indicators such as the maturity ogive, the spawning potential ratio, or the spawning stock biomass.

Until recently, the description of maturity phases for fish species sampled during annual scientific campaigns was mainly based on macroscopic criteria: the gonads of fish were extracted to allow a visual appreciation of criteria linked to reproductive status. For example, the length of the gonad, its color or global aspect, and texture were determining factors in maturity staging. Several National and European protocols were established by the International Council for the Exploration of the Sea (ICES) in order to harmonize this reading procedure. This harmonization led to the identification and description of six maturity phases named A to F (some of them subdivided into subphases): immature (A), developing (B), spawning (C), regressing/regenerating (D), omitted spawning (E), and abnormal (F) (Table 1). These phases, subphases, and their associated macroscopic criteria have been used since 2012 to homogenize the procedure of maturity evaluation for all teleost fish species regulated by European policies in stock management [4,5].

Even if this approach is simple and time-efficient, the results it yields are not entirely satisfactory. The assessment of visual criteria largely depends on the reader, which may lead to wide variability in the determination of the percentage of mature females. Indeed, the range of variability may reach 30% to 50% depending on the species and mainly concerns the first maturity phase (transition from immature A to first maturity B individuals) [5,7]. Subsequently, this variability affects the fine comprehension of the mechanisms controlling life history traits [8]. Moreover, the misevaluation of an individual’s maturity phase questions the quality of fisheries management procedures, negatively affects the modeling of stocks, and subsequently impacts management policies at a European scale.

A previous study [6] suggested combining morphologic criteria with histological criteria to accurately determine the reproductive phase of fish. Moreover, they reviewed the definitions and terminologies used in the description of reproductive processes occurring at the gonad’s level and at the cellular level. The different reproductive phases were associated with fine histological descriptions of the gonads. The classification of a female individual within a maturity phase relies on the identification of various stages of germinal cells, from oogonia to hydrated oocytes, but also the thickness of the ovarian wall, the occurrence of atresia, the number and aspect of blood vessels, and the presence of post-ovulatory follicles [9]. Indeed, a standardization of the histological description of each cell of the germ lineage was suggested so as to cover a large range of species based on the compilation of several terminologies [9]. For all species, the same cellular progression was observed during oogenesis: oogonia, primary growth oocytes, oocytes with cortical alveoli vesicles, vitellogenic oocytes, maturating oocytes, including the migration of the germinal vesicle and the hydration of the oocyte, and finally ovulation leading to the formation of post-ovulatory follicles. The histological approach, while expensive and time-consuming, is also considered to be the most accurate method to assess the reproductive phase of a species [10]. 

However, these descriptions are still based on a qualitative appreciation of histological structures, such as the presence of cortical alveoli oocytes or post-ovulatory follicles [11], the ovarian wall thickness, or even blood vessel abundance, and could greatly benefit from a quantitative approach. Moreover, since no norms are set concerning the studied surface of the histological section, observations may not be representative of the gonad section or of the whole gonad. In addition, if the course of gametogenesis is broadly unchanged in fish species, some species-specific features may be observed, such as the timing of certain cellular events or the morphology and proportions of each cell type. Therefore, a fine description of cellular characteristics for each species, combined with the use of a quantitative approach, would lead to a precise and accurate evaluation of microscopic criteria and subsequently the reproductive phase.

The pleuronectid *Pleuronectes platessa* is a Pleuronectiformes species intensively fished in the eastern English Channel, especially in **ICES** Area 27, and is subject to stock management procedures [12,13,14]. Indeed, the age and length at which 50% of females have reached sexual maturity have significantly shifted during the 20th century [15]. This further underlines the necessity to regularly monitor the evolution of maturity characteristics [16]. To complete the already existing works investigating fecundity and sexual maturity on the basis of macroscopic parameters, we offer in this study a precise and comparative description of oogenesis processes using Brown Peterson et al.’s (2011) [6] terminologies. We then developed a quantitative approach based on stereology and image analysis of the whole transverse ovarian section as an objective method to define the various phases, from the early immature phase to mature spawning females.

## 2. Materials and Methods

### 2.1. Specimen Sampling

Specimens of European Plaice (*Pleuronectes platessa*) were fished in the Eastern English Channel (**ICES** division VIId) (Figure 1) using a bottom trawl from either commercial fishing vessels or during the scientific campaign **CGFS** (Channel Ground Fish Survey). Individuals from campaigns were dead by the time of sampling, and all individuals from commercial catches were dead when received from the fish wholesaler. The sampling was opportunistic, depending on the monthly catches of commercial fishing vessels or the availability of scientific campaigns. This led to disparities in the number of fish sampled each month as well as gaps between sampled months. Moreover, commercial catches being under size regulations, small individuals could only be sampled during scientific campaigns that were themselves limited in time. The temporal frame for the samples was set following the reproductive cycle of the European Plaice in this area. Depending on commercial catch availabilities, a dozen fish were sampled per event, with double the samples taken during the spawning period. A total of 151 plaice were obtained from January 2017 to August 2019. All specimens sampled during the scientific campaign **CGFS** were treated immediately on board for biometric measurements (ungutted weight, total length, otolith extraction) [17]. All specimens from commercial vessels had been fished a day prior and were dissected as soon as they arrived on site. For each animal, the visceral cavity was opened to determine the sex, and the ovaries of females were carefully extracted, photographed, and weighted. From the gonadal weights, an individual gonadosomatic index (**GSI**) was computed using Equation (1).
GSI = (weight of the ventral ovary ÷ ungutted ovary-free weight of the individual) × 100(1)

The reproductive phase was established according to the **ICES** scale [5] through the observation of macroscopic criteria (Table 1).

### 2.2. Light Microscopy

The ventral ovary was transversely cut in the median region to obtain 1 cm-thick pieces of gonad for each female. The tissue samples were fixed for histology in Davidson’s fixative (10% glycerol, 20% formaldehyde, 30% ethanol 95°, 30% sterile seawater, 10% acetic acid) at 4 °C for 48 h. They were then dehydrated in successive ethanol dilution baths, transferred to butanol (Carlo Erba, Val-de-Reuil, France), and embedded in paraffin wax (Roth, Lagny-sur-Marne, France). Five micrometer sections were stained according to the Prenant Gabe trichrome protocol [18]. 

Microphotographs were taken with an Olympus AX70 microscope using the Olympus CellSens© software (v2.3). Various cell types of the female germline were identified, and measurements of cellular structure sizes (i.e., mean cell diameter and mean *zona pellucida* width) were made.

### 2.3. Quantitative Analysis of Ovarian Sections

The procedure for quantitative analysis was previously described [19]. Using a histology slide scanner (Aperio CS, Scan Scope Console software, v.10.2.0.2352), the slides were scanned and analyzed with the Aperio software (v12.1.0.5029). To quantify the cellular structures, a stereological analysis based on Glagolev’s [20] method was used. After detecting the cross sections on the scanned histological slides through segmentation using a threshold applied to the gray level image to extract the tissue area [21], a grid of 500 to 600 equidistant points was layered over the ovarian section using a random starting point. Each sampling point was assigned a single structure. This allowed for the computation of a percentage of times a structure was counted on a single slide, leading to a surface percentage for all ovarian structures. In this study, only the percentage of germ cell surfaces (**%GCS**) was taken into account for **og**, **po1**, **po2**, **cap**, **vit**, **pho**, **ho,** and **POF** (for the acronyms’ definitions see Table 2). To summarize the different follicle stages, **og**, **po1,** and **po2** were classified under **pca** (precortical alveoli), and all vitellogenic oocyte categories (**vit1**, **vit2**, and **vit3**) were classified under **vit**. With this data, information on the germ cells present in the ovaries of each individual, as well as the mean **%GCS** per month and for each maturity phase, was established. Inter-agent calibration with a reading protocol [22] was used to validate this stereological approach. Cellular homogeneity inter-gonad (ventral and dorsal) and intra-gonad (anterior median and posterior sections) was also checked [19]. Once cellular homogeneity was demonstrated, the reading results of the ventral ovary were used.

### 2.4. Size Class in Which 50% of Females are Mature (L_50_)

The size class in which 50% of the female population is considered mature (L_50_) was computed using a logistic regression between the maturity phase and the size using the function gonad_mature() of the SizeMat R package [23]. The L_50_ median and confidence intervals were estimated using a non-parametric bootstrap method with 1000 replications. The goodness-of-fit for the regression was evaluated using Nagelkerke’s R-squared coefficient of discrimination [24]. The L_50_ assumptions describe the range in which maturity onset occurs, represented by a logistic curve centered on length [25]. This curve indicates the point at which 50% of individuals start producing gametes relative to their body length, corresponding to the individuals presenting vitellogenesis. The data used to estimate this (L_50_) was the maturity phase determined through stereology by implementing histological features previously described [4,5,6].

## 3. Results

Ovaries appeared as paired, elongated structures within the abdominal cavity (Figure 2a). The gonadal section presents a concentric organization of the folded germinal epithelium that constitutes the ovarian lamellae, connecting to the external ovarian wall. This germinal epithelium contains the oogonia, oocytes, and/or follicles, depending on the stages of gametogenesis, until the release of mature oocytes from their follicles into the ovarian cavity just before spawning (Figure 2b,c).

Oogenesis was divided into four instances: proliferation, previtellogenesis, vitellogenesis, and maturation.

### 3.1. Proliferation

Oogonia are nested in the germinal epithelium. They consist of small oval cells (mean diameter 14.3 µm ± 4.6 µm) with a large round nucleus containing a unique nucleolus (Figure 3a). The nucleus contains clumps of chromatin in its periphery, and the cytoplasm is clear. The nuclear cytoplasmic ratio of an oogonium, calculated with Equation (2), is elevated (approximately 0.8). The nest usually groups several oogonia, accompanied by their associated somatic cells (elongated nuclei). Mitotic cell divisions are sometimes observed.
**N/C** = diameter of nucleus ÷ diameter of the follicle taken by the cytoplasm(2)

### 3.2. Previtellogenesis

Oogonia become primary oocytes following an increase in size (Figure 3b) as their cytoplasm becomes intensively hematoxylin-stained. A primary growth oocyte’s nucleus contains an increasing number of highly basophilic nucleoli, and their **N/C** decreases progressively. The nucleus of early-stage primary oocytes is spherical and surrounded by a smooth nuclear membrane. At this stage, the nucleus contains a few intensely red-stained nucleoli (less than 5 in general). Early-stage primary oocytes have a **N/C** of 0.5 or more and a mean size of 29.6 µm (±5.4 µm). At the following stage, the cytoplasm of advanced primary oocytes enlarges considerably, reaching a **N/C** of less than 0.5, and the nuclear membrane is still smooth. The number of nucleoli increases to more than 10, and they are located on the external periphery of the nucleus. The mean size of **po2** is 82.7 µm (±15.9 µm). At the end of this stage, it is sometimes possible to distinguish lampbrush chromosomes. 

At the end of the primary oocyte growth stage (Figure 3c), while the nucleus membrane is still smooth, the cytoplasm appears divided into two distinct concentric areas: the inner cytoplasm and the outer cytoplasm. At this stage, both areas appeared homogeneously stained without any clear vesicles, and the nuclear membrane was smooth. 

The cortical alveoli oocyte (**cao**) is characterized by the scalloping of the nuclear membrane and the presence of inclusions in the cytoplasm (Figure 3d). The number of nucleoli is still elevated, and they are niched into the folds of the scalloped nuclear membrane. Some scarce and small lipid droplets and cortical alveoli are restricted to the inner cytoplasm. Lampbrush chromosomes may be seen in the nucleus. During that stage, somatic cells forming the follicular cell layer and the theca, containing blood vessels, are clearly visible around the oocytes. The *zona pellucida* starts to form. The mean **cao** size is 153.1 µm (±17.8 µm).

### 3.3. Vitellogenesis

This stage is characterized by a significant increase in oocyte size linked to the accumulation of vitellus in the cytoplasm (Figure 4). At the beginning of the vitellogenic stage (**vit1**) (Figure 4a), a ring of eosinophilic vesicles containing vitellus forms around the cytoplasm of the oocytes. Some scarce lipidic droplets are still present. The nucleus is central, and its membrane remains indented. The *zona pellucida* slightly increases in thickness but remains green-stained. The mean size of **vit1** is 191.5 µm (±29.1 µm).

The stage named vitellogenic oocytes 2 (**vit2**) is characterized by the progressive invasion of the whole cytoplasm by vitellus droplets (Figure 4c). This progression starts from the external side of the oocyte until the vitellus occupies the whole cytoplasm. At this stage, the *zona pellucida* becomes eosinophilic as it progressively thickens, reaching a mean value of 9.2 µm (±1.6 µm). The mean size of **vit2** is 380.0 µm (±57.2 µm).

At the end of vitellogenesis, the oocytes (**vit3**) (Figure 4e) are completely filled with vitellus droplets. The follicles appear polygonal in shape, surrounded by a thick and fully eosinophilic *zona pellucida* (53.81 μm ± 5.10 μm). Vitellogenic oocyte **vit3** size reaches 629.5 µm (±88.5 µm).

### 3.4. Maturation

During the hydrating stage, partially hydrated oocytes contain vitellus droplets but also homogenous, clear basophilic areas in their cytoplasm (Figure 5a). There is a slight decrease in *zona pellucida* thickness, which appears striated (33.1 μm ± 3.5 μm) with an increase in the mean diameter of oocyte follicles (847.2 µm ± 104.0 µm).

At the end of the hydration process, hydrated oocytes appear spherical (Figure 5b). A fully hydrated oocyte (**ho**) possesses a homogeneous cytoplasm with no visible vitellus droplets. The mean diameter of **ho** is 958.7 μm (±60.7 μm), while *zona pellucida* thickness reaches 40.4 μm (±5.0 μm). Figure 6 illustrates female germinal cells’ growth diameter from the oogonia stage to mature oocytes. Moreover, *zona pellucida* thickness increases mainly between **vit2** and **vit3** stages. The presence of follicle cells and theca indicates that the oocyte has not been emitted yet. The observation of well-preserved, hydrated oocytes is tricky due to the difficulty of cutting at this stage.

### 3.5. Other Notable Ovarian Structures

Post-ovulatory follicles are observed in the ovary once the oocytes have been emitted (Figure 5c). This structure is composed of theca and granulosa layers left behind after ovulation. Due to ongoing lysis, the thickness of these layers will progressively diminish over time during the regressing phase (D).

Atretic oocytes were observed on histological slides of individuals at different maturity phases (Figure 5e,f). Atretic oocytes may be classified into two main categories: Early atretic oocytes, or atretic oocyte alpha, exhibit evidence of lysis before emission. They are still enclosed in the theca and granulosa but show evidence of lysis, resulting in a progressive disruption and the loss of their round shape. Late atretic oocytes, named atretic oocytes beta, are emitted oocytes remaining in the lumen of the ovary, out of their post-ovulatory follicle. They are always free of theca and surrounded by phagocytic cells. Areas with lysis were also observed without any possibility of determining the nature of the lysed cells (Figure 5d). During maturation, the ovarian wall appears distended due to the presence of **ho** and progressively thickens with abundant muscular fibers and blood vessels after spawning. All these cellular structures have been taken into account during the quantitative approach.

### 3.6. Quantitative Evaluation of Maturity

Stereology allows for the count of structures found within a 3-dimensional cell by analyzing a 2-dimensional section while taking into consideration geometry and statistical probabilities [26,27]. As applied to histology, linking a single cellular structure to a unique sampling point gives an objective view of the different cellular structures present within the cross section. This provides a means to quantitatively evaluate the surface percentage of these structures or to methodically point out individuals presenting specific cellular stages throughout the studied cross section.

Through this method, Figure 7 shows for fish of each size class the percentage of animals classified into the most advanced oocyte stage present within the ovary. When taking into account precortical alveoli oocytes (i.e., **og**, **po1,** or **po2**) as the most advanced follicle stage, all individuals of 18 cm or less are classified into that category. The first individual with **cao** as the most advanced oocyte stage appears at 19 cm (n = 1 out of 3 sampled individuals). Individuals presenting vitellogenesis (i.e., **vit1** and later stages) as the most advanced oocyte stage are 21 cm and over. 

Figure 8a shows the percentage of mature individuals with their most advanced germline stage, classified by month over a year (March 2018 to March 2019). All individuals (n = 123) are ≥21 cm long and could be considered sexually mature when looking at Figure 7. The first sampling in March 2018 showed a majority (80%) of individuals presenting **POF**, an indication of spawning. In June, 58% of individuals showed **pca** as their most advanced oocyte stage, while the rest showed **cao**, indicating that spawning events ended between March and June. From November to December, the number of individuals with partially hydrated oocytes at the most advanced stage increases. In January and February, the majority of individuals (70% and 91%, respectively) presented **POF**, signaling that spawning has occurred. By March 2019, individuals with **POF** were still present (46%), but there are still individuals with **pca** (3%), **cao** (27%), vitellogenic oocytes (17%), and **ho** (7%) as their most advanced oocyte stage, indicating oogenesis onset. 

Furthering the use of histology data obtained through stereology, Figure 8b was established to allow for a better appreciation of the mean surface percentages of each germ cell type (**%GCS**) found within the ovaries depending on the month. The mean percentage of precortical alveoli (**pca**) oocytes ranges from 31% in December to 56% in March 2019, and they are present all year round. Cortical alveoli stages also seem to be present all year, with a mean surface percentage of 22% in November that decreases to 10% by January. For November and December, the mean percentage of maturing oocytes (**vit**, **pho,** and **ho**) increased from 24% to 48%. By January, **ho** are present (5%) concomitantly with **POF** (38%), indicating a massive spawning event. From January to March, **POF** progressively decreased due to its progressive resorption. 

The gonadosomatic index (**GSI**) was established for each individual classified under a visually determined maturity stage A, B, C, or D (Figure 9a and Table 1) for all sampled animals (n = 151). The mean **GSI** shows the highest rates (22%) for individuals classified under C (spawning) (n = 3) and the lowest mean **GSI** (<1%) for phase A (immature) (n = 49). When using quantitative data issued from stereologic analysis (Figure 9b,c), immature A animals were mainly associated with **pca** or **cao** as the most advanced stage (94%), whereas 6% of them exhibited **POF**, attesting that they had already spawned. On a stereological level, however, **POF** represented only about 2% of the total germ cell surface. For developing animals (B) (n = 53), approximately 43% had **POF** as the most advanced stage, even if they represented only 9% of the total germ cell surface. For spawning animals (C), 66% have **ho** or **POF** as the most advanced stage. Maturity phase A shows a majority of **pca** stages (80%), but the presence of **cao** (17%), **ho** (<1%), and **POF** (2%) underlines misclassifications. This problem is also found in phase B, with 76% of germ cells that are of **vit** stage and under, but **pho** (12%), **ho** (3%), and **POF** (9%) are present. Phase C shows a majority of **pho** (70%), with no **cao** stage or below, but the presence of **vit** (14%), **ho** (15%), and a few **POF** (1%). Finally, for phase D, all germ cell types are present, with the highest mean surface percentage for **pca** (34%), followed by **POF** (25%), and **vit** (22%).

The size class in which 50% of the females in our sampling are considered mature by stereology (L_50_) is presented in Figure 10. The L_50_ is 20.6 cm, with a confidence interval of [19.6, 21.5] cm and a Nagelkerke’s R-squared of 0.7.

## 4. Discussion

The minimum landing size in fisheries management is established through sound knowledge of age and size at maturity for each stock species considered. The present study details the oogenesis cycle for *Pleuronectes platessa*, a species with an age and length at maturation that has been in decline for the past century in the North Sea [16,28]. Yearly surveys and samplings from commercial catch allowed for important data collection at different time periods, making it possible to estimate an age or length at first maturity for this species. The aim of this study was to establish an accurate description of the oogenesis cycle for this pleuronectid as well as underline the main criteria linked to the spawning period of mature individuals. For this, a histological approach, enhanced through the use of stereology, was used [19].

### 4.1. Germ Cell Lineage

Before quantifying cell structures through stereology, a detailed description of the *Pleuronectes platessa* oogenesis cycle needed to be implemented since gametogenesis descriptions for the European Plaice are scarce and dated [29,30,31,32,33,34,35]. *Pleuronectes platessa* is an iteroparous batch spawner with group-synchronous oocyte development and a determinate fecundity [36]. This batch spawner will release hydrated oocytes at two- to five-day intervals for four to six weeks in the North Sea [37]. Even if all cellular structures found within the female gonad were identified, only germline cells were detailed in this paper. Notable characteristics were found for European Plaice oogenesis regarding lipid droplet abundance, *zona pellucida* size, and hydrated oocyte size.

### 4.2. Lipid Droplets

In teleosts, lipids found in eggs play an important role in oocyte maturation, egg viability, and embryonic and larval fish growth [38], but they also influence the time intervals between spawning and egg hatching [39] and can vary in size and presence depending on the species [40]. In a previous study, the volume percentage of oil globule(s) within the oocyte was reviewed for 803 teleost species, including *Pleuronectes platessa* [41]. For Pleuronectiformes, the volume percentage of oil globules within the oocyte seems low (under 2.5%), as well as for Pleuronectidae (given at 0% for the European Plaice). This concurs with our observations, with very small and scarce lipid droplets appearing during the cao stage. These lipid droplets will be in very low abundance throughout the entire oogenesis cycle, compared to other flatfish species such as *Solea solea* or *Lepidorhombus* spp. (based on personal observations). 

With proteins, lipids are the oocytes’ main energy source in vitellus. Lipid and protein contents in ovaries differ in Pleuronectiformes [42]. Indeed, *Pleuronectes platessa* showed low ovarian lipid content (8%) with high proteins (85%), compared to other flatfishes [43]. For the European Plaice, these differences in lipid within the ovary correlate with the low presence of lipid droplets in oocytes.

### 4.3. Zona Pellucida Size

The *zona pellucida* (**zp**) envelopes the oocytes and is a crucial structure not only during oogenesis but also during ovulation, fertilization, and embryonic development [44,45,46]. Moreover, it is one of the embryonic structures that will play an important role in protecting the embryos by mitigating the effects of potentially harmful factors such as high temperatures, salinity, anoxia, and desiccation [47]. In this study, the first appearance of **zp** is noted at the cortical alveoli stage (**cao**) and progressively thickens through the stages. Similar observations have been recorded for the European Plaice [35,48]. However, this study goes further with **zp** sizes and descriptions for undischarged oocytes, with the first appearance of a **zp** in **cao**. The **zp** progressively thickens throughout the vitellogenesis process until reaching a maximum size of 53.81 μm (±5.10 μm) at the late vitellogenic stage. Its composition also changes during this stage, as indicated by the evolution of the staining. The **zp** will then lose thickness during the hydration process and reach a size of 40.41 μm (±5.00 μm) for undischarged hydrated oocytes. Such a progressive reduction in **zp** thickness during hydration was already observed in various fish species [49]. Once ovulation has occurred, the **zp** will decrease further in size [50], with pelagic eggs’ envelopes being much thinner than those of demersal eggs [51,52]. However, it has been previously found that the European Plaice’s pelagic eggs possess an exceptionally thick **zp** of about 15 μm [51].

### 4.4. Hydrated Oocyte Size

In this study, the hydrated oocyte mean size prior to ovulation was 958.66 μm (±60.66 μm). In previous studies, North Sea Plaice egg diameter ranged from 1.66 to 2.17 mm [53], and fresh egg mean size ranged from 1.736 to 1.966 mm for samples collected from 1984 to 1986 [54]. Other follicle sizes were measured for the Irish Sea Plaice, with an estimated follicle diameter ranging from 1.65 to 1.80 mm for a 43 cm female [55]. Another mean follicle size of 592 μm was measured for one- to two-year-old European Plaice, and a 789 μm mean size was found for individuals four years of age and older [56]. The smaller size estimated in this study could be explained by different factors. First of all, the dehydrating tissue processing method used may lead to smaller-sized oocytes. Secondly, the mean diameter was estimated through the widest diameter of sectioned oocytes, but there were no guarantees that all sections were fully transversal. Moreover, the measures (n = 22) were from a small-sized (21 cm for 70 g) individual sampled in 2017 that had already spawned (presence of post-ovulatory follicles). These are important facts since it has been shown that larger females often produce larger eggs [56,57], and egg size progressively decreases with each batch during the spawning season [57].

### 4.5. Histology for Maturity Staging

Compared to estimating sexual maturity visually through macroscopic criteria, histology was found to yield more accurate results as well as more information [10,58,59,60,61]. Adding the quantitative approach (stereology) allows for a quantification of the different structures as well as estimating their representativeness within the ovary. Even with the use of histology, it can be very difficult to distinguish immature animals from mature animals that are capable of producing hydrated oocytes but are in the resting phase if the samplings are conducted outside of the spawning period. With stereology, monthly samplings allow for the quantification of each cellular type along the oogenesis cycle. Add to that different-sized fish, and it is possible to determine size (or age) at first maturity and eventually see if there are differences in spawning times for fish of different sizes.

### 4.6. Size at Maturity

Identification of cortical alveoli oocytes (**cao**) is a key point in maturity staging since their formation is gonadotropin-dependent [62,63], placing them into the secondary growth oocyte category [6,63]. When distinguishing between fish with precortical alveoli oocytes, **cao**, or vit as their most advanced oocyte stages, the minimum size varies. The appearance of cao will be the turning point between a fish classified as sexually immature and a mature one. When classifying individuals of different sizes with cao as the most advanced stage, the first size at maturity is 19 cm. When using the presence of vit as the most advanced stage, the size at first maturity is 21 cm. The size at which all individuals are sexually mature is 23 cm. This, however, could be biased when taking into account the constrained sampling size as well as the sampling period. Indeed, a sampling of the spawning period could explain the presence of large individuals (24 cm and over) with precortical alveoli oocytes as the most advanced follicle stage. Moreover, increasing the sampling for small-sized individuals (between 18 cm and 24 cm) would help better define the size limit. This would also better underline the individual variability in size at first maturity for this stock.

The estimation of the size class in which 50% of the females of our sampling (L_50_) are considered mature by stereology was 20.6 cm, very close to the minimum size of the individuals presenting vitellogenesis (21 cm). These results are well in accordance with the L_50_ calculation assumptions.

When quantifying the structure with stereology, the identification of **cao** was not always obvious, with cortical alveoli being scarce and small-sized, leading to a probable underestimation of this key germ cell. An alternative option would be to set the limit between the mature and immature states with the presence of **vit**, but as stated previously, this would lead to a difference of 2 cm for the size at first maturity. The already low size at first maturity found in this study follows the decrease in age and length at maturation that has been occurring for the past century for the European Plaice, at least since 1955 [16]. A previous study [28] showed that the Lp50 (probability of becoming mature is 50%) of four-year-old female *Pleuronectes platessa* decreased from 41.6 cm in the 1900s to 33.3 cm in the 1980s and 22.9 cm in the 2000s. The observed changes in maturation, reproductive investment, and growth seemed consistent with fisheries-induced evolution [16]. Moreover, with the current context of global changes, the evolution and availability of trophic resources could further impact the age and/or size at first maturity for this species [2]. These observations strongly underline the importance of using histology at such a fine level in order to better follow the evolution of life history traits in fish and help with fishing management regulations.

### 4.7. Spawning Period

Classifying the sampled individuals by the most advanced germline cell present within the ovary, by month, allowed for an illustration of the yearly cellular cycle. Precortical alveoli germ cells are present all year long and in individuals of all maturity phases. Partially hydrated oocytes are present by November for a few individuals. The first spawning takes place between December (presence of **pho**) and January (presence of **POF**). The majority of individuals had been spawned by February. This concurs with the literature stating that the spawning period for female North Sea Plaice starts in December [2] and January [37], with a peak in February [2,37]. From January to March, an increase in mature oocytes (**vit** and **ho**) may suggest another possible batch spawn by April, such as that observed by Rijnsdorp [37]. However, we observe an increase in **vit** stage oocytes from March to June, while Rijnsdorp [37] suggested a complete renewal of the oocytes in the ovary after the expulsion of hydrated oocytes, with a start of vitellogenesis in July and a marked gonadal growth by October.

### 4.8. Maturity Phase

The gonadosomatic index (**GSI**) clearly distinguishes female individuals classified under phase C (spawning) from other maturity phases, though it is important to note that only three individuals had been sampled for this phase. The low number of histological slides for spawning (phase C) individuals can be explained by the difficulty in obtaining exploitable cross sections from ovaries with fragile, hydrated oocytes. Phase C is considered the easiest phase to identify with the naked eye because of the presence of hydrated oocytes that can reach a diameter of more than 1 mm. Value deviations in the **GSI** for phase B can be explained by the presence of visually misclassified fish. Indeed, when looking at the histological slides, fish with the highest **GSI** classified under phase B show late-stage vitellogenic oocytes and oocytes in hydration in their ovaries. 

Other misclassifications are present between phases A (immature) and D (regressing/regenerating), with individuals that have spawned classified under phase A, implying that mature and spawning-capable individuals have been classified as immature individuals. An additional misclassification is found between B and D, with 42% of individuals that have spawned being classified under phase B. 

The difficulty of visually determining a maturity phase for the European Plaice has already been addressed [64], with the highest staging disagreements between immature (A) and developing (B) females. Finally, a maturity exchange workshop for *Pleuronectes platessa* [5] showed the lowest agreement of 67% for phase D and the highest agreement of 83% for phase C. The use of histology gives a more detailed resolution for maturity phase profiles compared to the **GSI** and visual maturity determination.

## 5. Conclusions

Spawning-capable female individuals have been identified using histology. From this, the issue of misclassification through the use of the visual maturity staging method for the European Plaice has been made clear. This allows us to validate the method of using histological criteria to classify individuals into a maturity phase. However, this study also underlined several difficulties, notably the limited number of certain individuals’ sizes due to the opportunistic samplings as well as the loss of histological sections due to the fragility of the ovaries. Indeed, an accurate determination of maturity data is of utmost importance in fisheries management, regardless of the time- and resource-consuming nature of histology. The use of histology for maturity determination is not a new method, but coupling histology with stereology to assess maturity brings great flexibility to the method and may be applied to other species. However, even if similarities are found among general germline cell stages, germline cells defining a particular maturity phase must be adapted from one species to another. The next step would be to standardize these procedures through the establishment of detailed species-specific gametogenesis lexicons, a stereology reading protocol, reader calibrations, and gonad homogeneity validations. This would allow for a more objective determination of the maturity phase. Furthermore, data collection at such a fine scale may lead to the possibility of automating the reading process through image analysis and supervised learning, allowing for the analysis of a higher number of samples in a shorter time period.

## Figures and Tables

**Figure 1 animals-13-02506-f001:**
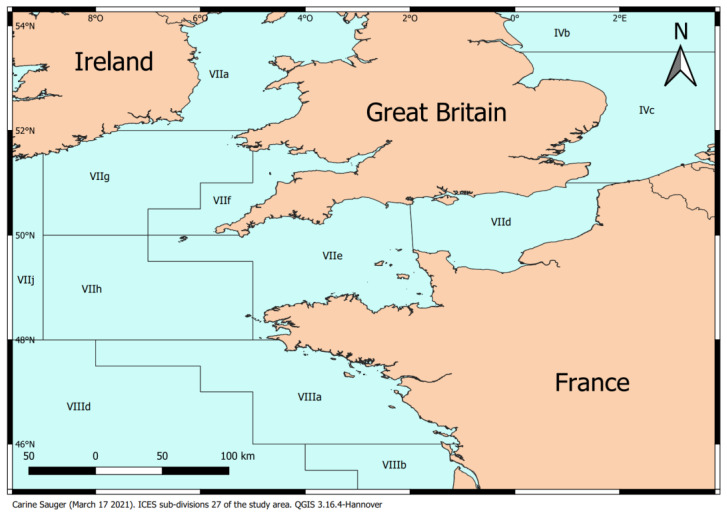
Map of **ICES** Area 27. The study took place in the VIId division. (Carine Sauger-March 17 2021-QGIS 3.16.4-Hannover).

**Figure 2 animals-13-02506-f002:**
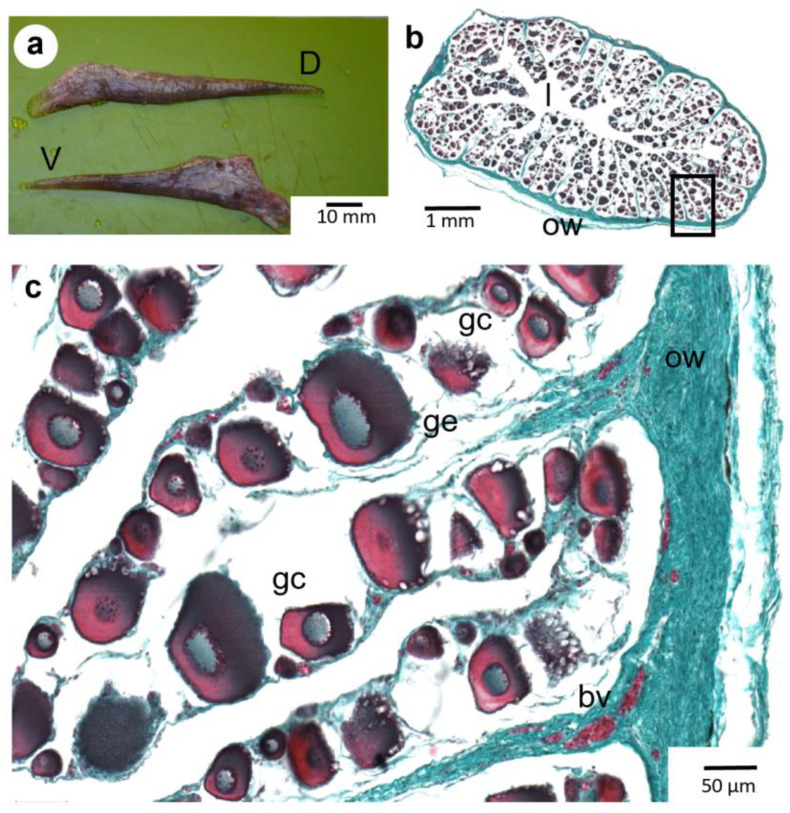
(**a**): Dorsal (D) and ventral (V) ovaries for *Pleuronectes platessa*. (**b**,**c**): Transverse histologic sections of *P. platessa* ovaries showing a concentric organization. Germinal cells (gc) at various stages of oogenesis are located between two successive germinal epithelium lamellae with (**c**): an enlargement of the insets in (**b**). ow: ovarian wall (ow); ge: germinative epithelium; l: lumen of the ovaries; bv: blood vessels.

**Figure 3 animals-13-02506-f003:**
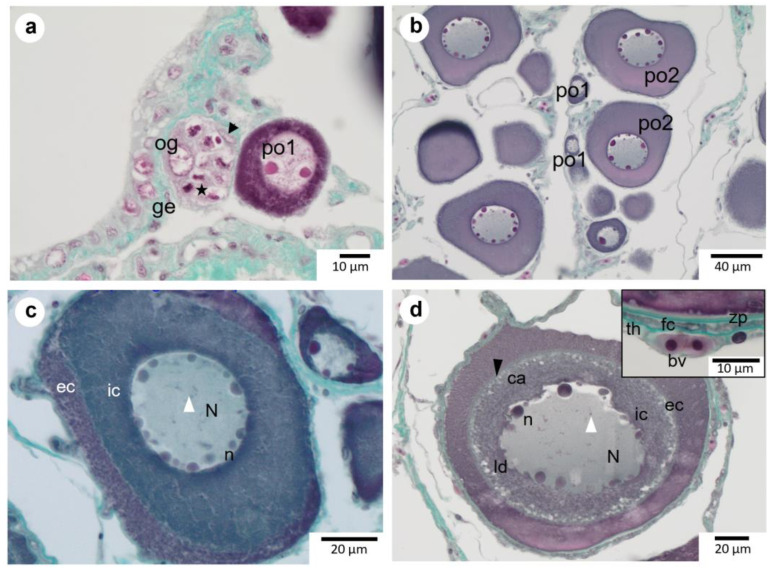
Oogonia, primary growth oocytes, and early vitellogenic oocytes (vit1) of *P*. *platessa*. (**a**): Oogonia (**og**) associated with somatic cells (black arrow) nested in germinative epithelium (**ge**). Note an oogonia nest with intense proliferative activity (★). (**b**): An early primary oocyte (**po1**) with a low number of nucleoli and highly stained cytoplasm and more advanced primary oocytes (**po2**) with abundant nucleoli (**n**) along the periphery of the nucleus (**N**). The nuclear membrane is still smooth. (**c**): Precortical alveoli stage, classified under **po2**, with cleaved **ic** and **ec** (internal and external cytoplasm, respectively). Lampbrush chromosomes (the white arrowhead) may be observed. (**d**): Cortical alveoli oocytes (**cao**) with a scalloped nuclear membrane. Lipidic droplets (**ld**) are observed in the inner cytoplasm. Cortical alveoli (**ca**) can be seen along the junction between the inner and outer cytoplasms. The follicle envelope is composed of a thin *zona pellucida* (**zp**) encompassed between the oocyte membrane and the complex follicular cells (**fc**) and theca (**th**) containing blood vessels (**bv**).

**Figure 4 animals-13-02506-f004:**
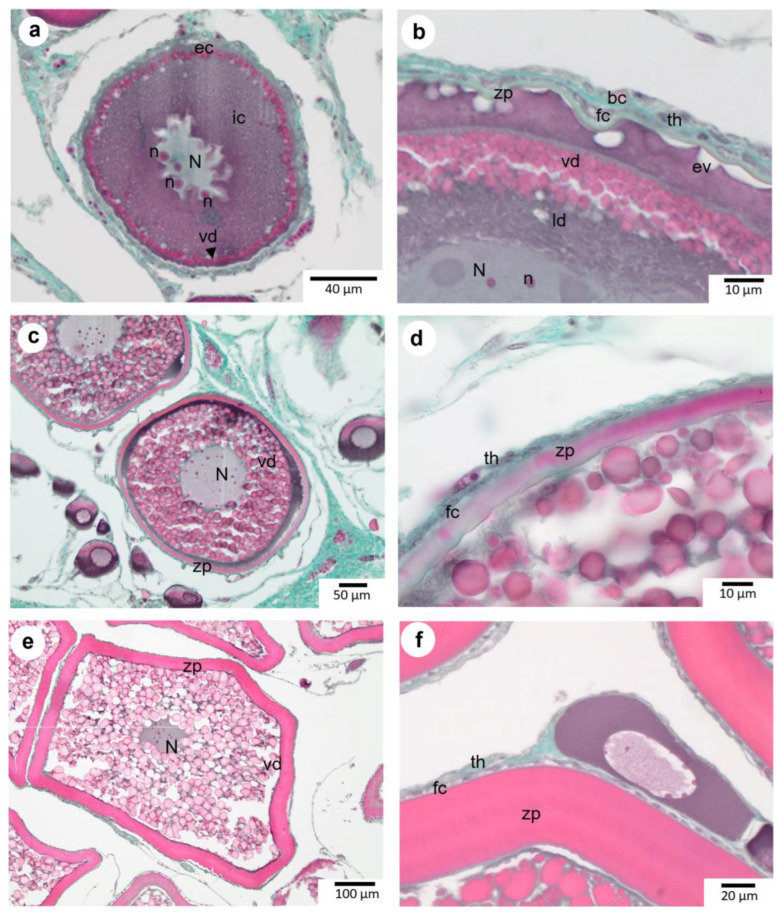
Vitellogenic oocytes of *P. platessa*. (**a**,**b**): Early vitellogenesis in **vit1** oocytes, showing the formation of pink-stained eosinophilic vitellus droplets (**vd** and black arrow), initiates the start of vitellogenesis. At this stage, vitellus droplets coexist with scarce lipidic droplets (**ld**). The cytoplasm is zonated (**ic** and **ec** for internal and external cytoplasm, respectively). The *zona pellucida* (**zp**) is still thin and basophilic. Endocytosis vesicles (**ev**) are clearly visible under the *zona pellucida*. The follicular cells (**fc**) and the theca (**th**) containing blood vessels (**bv**) encircle the oocyte. (**c**,**d**): Mid vitellogenic oocyte (**vit2**) showing progressive invasion of the cytoplasm by vitellus droplets. The **zp** progressively thickens and becomes eosinophilic. (**e**,**f**): Late vitellogenic oocyte (**vit3**) with the thickening of the **zp** fully red and vitellus droplets filling the entirety of the cytoplasm. **N**: nucleus; **n**: nucleolus.

**Figure 5 animals-13-02506-f005:**
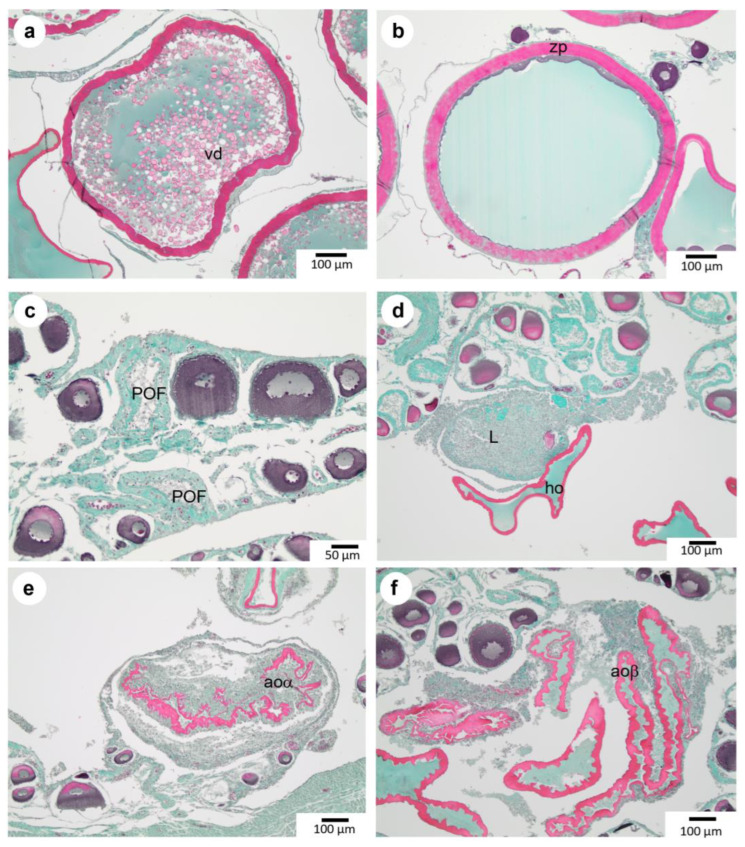
Hydration of oocytes, emission, and atresia in the ovary of *P. platessa*. (**a**): Partially hydrated oocytes (**pho**). (**b**): Hydrated oocytes (**ho**). Other structures observed in the ovary are (**c**): post-ovulatory follicles (**POF**) remaining in the ovary after ovulation. (**d**): Lysis area (**L**) in the lumen. (**e**): Atretic oocyte exhibiting evidence of lysis before emission (**aoα**). (**f**): Atretic oocyte exhibiting evidence of lysis after emission (**aoβ**).

**Figure 6 animals-13-02506-f006:**
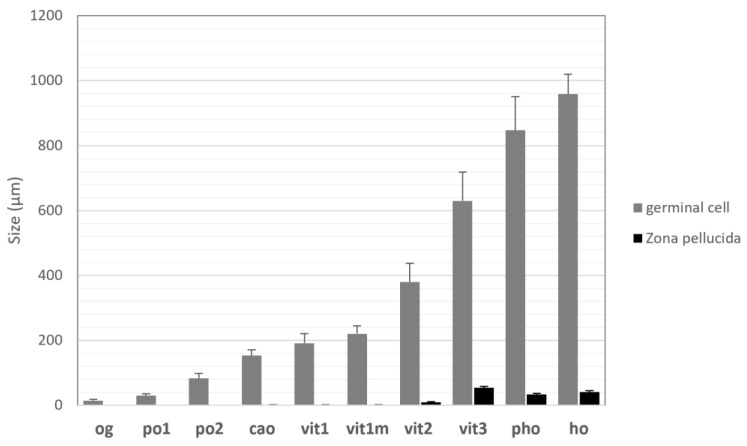
Mean diameters (µm) of female germline cells and *zona pellucida* (**zp**) thickness for *P. platessa* during oogenesis. Mean diameters were measured over 20 cell sections, including the nucleus. Bars represent the standard deviation. **og**: oogonia; **po1**: early primary oocyte; **po2**: advanced primary oocyte; **cao**: cortical alveoli oocyte; **vit1**: early vitellogenic oocyte; **vit2**: advanced vitellogenic oocyte; **vit3**: complete vitellogenic oocyte; **pho:** partially hydrated oocyte; **ho**: hydrated oocyte.

**Figure 7 animals-13-02506-f007:**
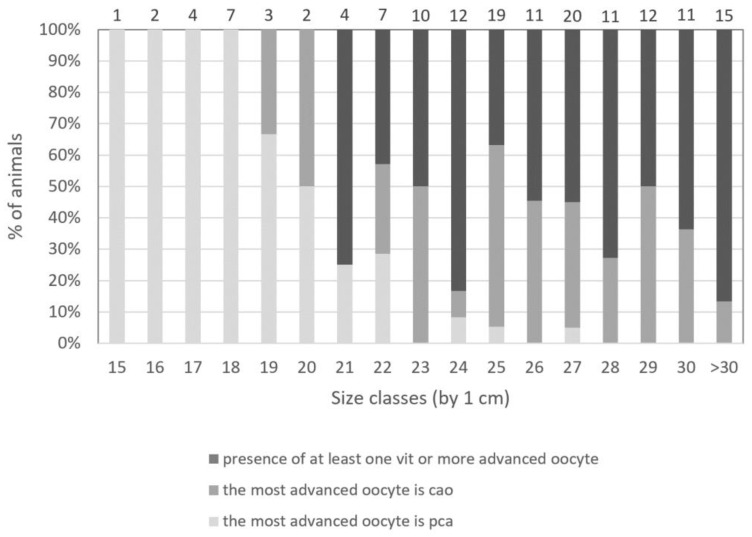
Identification of the most advanced oocyte in the gonad of *P. platessa*. For each individual’s length (by class of 1 cm), three groups were defined: animals with at least one vitellogenic oocyte or more advanced stages (dark gray), animals with at least one cortical alveoli oocyte (**cao**) as the most advanced oocyte (medium gray), and animals with precortical alveoli (**pca**) stages (**og** + **po1** + **po2**) as the most advanced oocyte (light gray). The number (n) of fish belonging to each size class is indicated on the top of the bars.

**Figure 8 animals-13-02506-f008:**
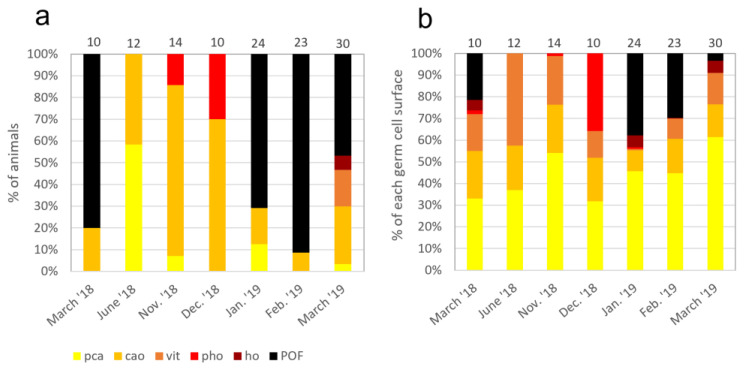
(**a**): For each month, percentage of animals classified under the most advanced oocyte stage found within their ovary. (**b**): For each month, the mean percentage of germline cell surface (**GCS**) found in the gonads of sampled *P. platessa*. **pca**: precortical alveoli oocytes (**og** + **po1** + **po2**); **cao**: cortical alveoli oocyte (light gray); **vit**: vitellogenic oocyte; **pho**: partially hydrated oocyte; **ho**: hydrated oocyte; **POF**: post-ovulatory follicle. The number of fish sampled (n) for each month is indicated on the top of each bar.

**Figure 9 animals-13-02506-f009:**
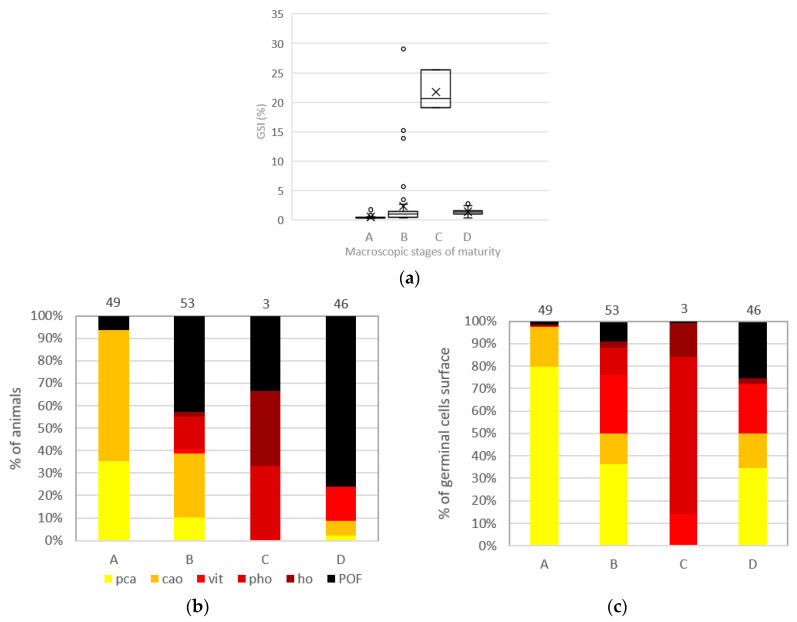
*Pleuronectes platessa* classified under a maturity phase through visual macroscopic criteria with (**a**) boxplot of gonadosomatic index (GSI %) with minimum, first quartile, median, third quartile, maximum. Black cross indicates the mean value. (**b**) percentage of animals by the most advanced oocyte stage found within their ovary and (**c**) mean % GCS (germline cell surface). With A: immature; B: developing; C: spawning; D: regressing/regenerating. Germline cells were identified through stereology and the percentages were computed from the total germ cell population in each maturity phase. pca: precortical alveoli oocytes (og+po1+po2); cao: cortical alveoli oocyte; vit: vitellogenic oocyte; pho: partially hydrated oocyte; ho: hydrated oocyte; POF: post-ovulatory follicle. The number of fish sampled for each month (n) is indicated on the top of each bar for (**b**,**c**).

**Figure 10 animals-13-02506-f010:**
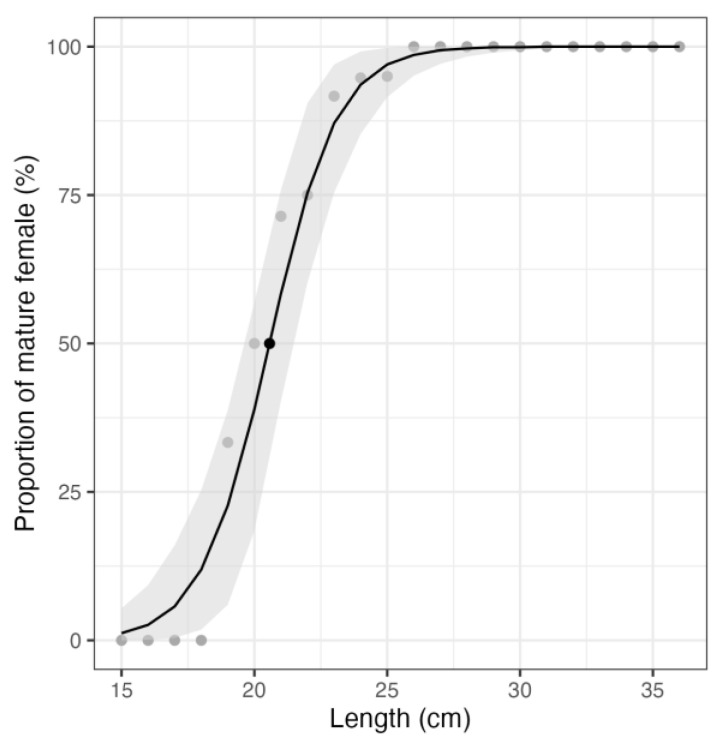
Logistic regression between the size (x-axis) and the proportion of mature females in the sampling (y-axis). The gray dots are the individual measurements, the black dot is the L_50_ (20.6 cm), and the gray ribbon displays the estimates’ confidence interval using a non-parametric bootstrap with 1000 replications. The confidence interval of L_50_ is [19.6, 21.5] cm.

**Table 1 animals-13-02506-t001:** Description of the maturity phases for female teleostean species adapted from the Workshop for Maturity Staging Chairs (WKMATCH), the Workshop for Advancing Sexual Maturity Staging in Fish (WKASMSF) [4,5] and Brown Peterson et al. (2011) [6].

State Description	Sexual Maturity Phase		Old Terminology (ICES 2012)		New Terminology (ICES 2018)		Macroscopic Criteria	Histological Features
		Possible Sub-Phase		Possible Sub-Phase		Possible Sub-Phase		
SI: Sexually immature (without gonad development)	Immature		I		A		Small, pinkish, and translucent (often clear) ovaries shorter than 1/3 of the body cavity. Indistinct blood vessels and no visible eggs	Only oogonia and primary growth oocytes were present, containing no oil droplets, rare atresia, and no muscle bundles. Thin ovarian wall, scarce connective tissue around follicles, and little space between oocytes
SM: Sexually mature (with gonad development)	Developing	Developing but functionally immature (first-time developer)	II	IIa	B	Ba	Enlarging ovaries, blood vessels become more distinct	Increase in oocyte size, blood vessels more distinct, primary growth, and cordical alveolar oocytes present. Early and mid-vitellogenesis oocytes and atresia may be present. No post-ovulatory oocytes nor late vitellogenic oocytes
		Developing but functionally mature		IIb		Bb		
	Spawning	Actively spawning	III	IIIa	C	Ca	Large ovaries, prominent blood vessels, and individual oocytes are visible macroscopically	Late vitellogenic oocytes are present. Early stage of oocyte maturation can be present. No post-ovulatory. Atresia of vitellogenic or hydrating oocytes may be present
		Spawning-capable		IIIb		Cb		Oocytes at the end of germinal vesicle migration, germinal vesicle breakdown, hydration, or ovulation are present. Recently collapsed post-ovulatory follicles can be present
	Regression/Regeneration	Regression	IV	VIa	D	Da	Reddish ovaries of about 1/2 of the body cavity length. Flaccid ovary walls, prominent blood vessels, and possible remnants of disintegrating opaque and/or translucent eggs	Atresia (any stage) and post-ovulatory oocytes are present. Some healthy cortical alveolar oocytes and/or early and mid-vitellogenic oocytes are present
		Regeneration		IVb		Db	Small, pinkish, and translucent ovaries that have a length of about 1/3 of the body cavity, with reduced but present blood vessels. No visible eggs	Only oogonia and primary growth oocytes are present. Oil droplets can be seen in primary growth oocytes (species-dependent). Muscle bundles, enlarged blood vessels, a thick ovarian wall, and/or atresia or old, degenerating post-ovulatory oocytes may be present
	Omitted spawning		V		E			No post-ovulatory oocyte and at least 50% of the vitellogenic oocytes are atretic
	Abnormal		VI		F		Problems in development (necrosis, sclerosis, intersex; the majority are unhealthy)	

**Table 2 animals-13-02506-t002:** Summary of used abbreviations, germline cell identification criteria, and mean cell size (µm) in *P. platessa*. **pca**: precortical alveoli; **vit**: vitellogenic oocyte categories.

Abbreviation		Name	Identification Criteria	Size (µm)
pca	og	Oogonia	N/C ≈ 0.8 Clear cytoplasm	14.3 ± 4.6
	po1	Primary oocyte, early stage	N/C ≥ 0.5. Hematoxylin-stained cytoplasm. Smooth nuclear membrane. Less than 5 nucleoli	29.6 ± 5.4
	po2	Primary oocyte, late stage	N/C < 0.5. Hematoxylin-stained cytoplasm. Smooth nuclear membrane. More than 10 nucleoli. At the end of the stage, 2 distinct cytoplasmic areas.	82.7 ± 15.9
cao		Cortical alveoli oocyte	Scalloping of the nuclear membrane. Numerous nucleoli. Scarce lipidic droplets and cortical alveoli in the inner cytoplasm. Lampbrush chromosomes. *Zona pellucida* starts to form.	153.1 ± 17.8
vit	vit1	Early vitellogenic oocyte	Ring of eosinophilic vesicles around the cytoplasm. Scarce lipidic droplets. Migration of the germinal vesicle. *Zona pellucida* thickness increases.	191.5 ± 29.1
	vit2	Mid vitellogenic oocyte	Progressive invasion of the whole cytoplasm with vitellus droplets. *Zona pellucida* becomes eosinophilic	380.0 ± 57.2
	vit3	Late vitellogenic oocyte	Cytoplasm completely filled with vitellus droplets. Polygonal in shape. Thick and fully eosinophilic *zona pellucida*	629.5 ± 88.5
pho		Partially hydrated oocyte	Presence of vitellus droplets and clear basophilic areas in the cytoplasm.	847.2 ± 104.0
ho		Hydrated oocyte	Homogeneous cytoplasm with no visible vitellus droplets	958.7 ± 60.7
POF		Post-ovulatory follicle	Theca and granulosa layers left behind after ovulation	

## Data Availability

All scanned histological slides as well as reading results are available on the Zenodo data repository (https://zenodo.org/record/3463296#.ZB115PaZOF4, accessed on 27 September 2019). For the scripts, contact the main author, C.S.

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
