# Peer review of "A Quantitative Histologic Analysis of Oogenesis in the Flatfish Species Pleuronectes platessa as a Tool for Fisheries Management"

_animals, 2023, doi:10.3390/ani13152506_

Round 1
Reviewer 1 Report (Previous Reviewer 2)
Dear Authors,
Thank you very much for considering my corrections and suggestions. After making corrections, I believe that the work is much clearer and legible.
Sincerely, R2
Author Response
Reply to the Editor:
In regards to the quality of the English used in the submitted article, all modifications may be
found highlighted in the text.
Reviewer 1:
- Presentation of the results can be improved
- Thank you very much for considering my corrections and suggestions. After making
corrections, I believe that the work is much clearer and legible.
Reply :
Thank you again for the previous suggestions and for reviewing this article favorably.
All modifications may be found highlighted in the text
Please see attachment

Reviewer 2 Report (New Reviewer)
The authors provide a good, interesting observation of maturity development on a such important species.
Althoug the manuscript apperars well-structured enough, some revisions need. Generally, I strongly suggest for summarizing the discussions. this section resulted too long, often including argument and hypotheses related only marginally to the main aims of paper.
Finally, some other comments are available in the pdf attached.

A moderate revision of English is kindly recommended.
Author Response
Reply to the Editor:
In regards to the quality of the English used in the submitted article, all modifications may be
found highlighted in the text.
Reviewer 2:
- Moderate editing of English language required
- Summarize the discussion : presence of arguments and hypotheses related only
marginally to the main aims of the paper.
- Keywords should not include words already in the title
- Add literature regarding maturity evaluation using gonad histology (commentaire dans
google drive pour plus d’info)
- Why are the acronyms in the manuscript in bold type?
- The author should redo the GSI estimates avoiding the use of total weight (commentaire
dans google drive pour plus d’info)
- Does "all sampled animals" mean the GSI has been estimated for both sexes?
- Figure 9 : Why was macroscopic maturity preferred to microscopic analysis?
- Move the L50 description in 4.6 to the M&M
Reply :
The authors wish to thank Reviewer 2 for their constructive criticisms and their attention to
detail.
The document was reviewed and a few spelling mistakes were corrected while all contractions
were removed.
Words already present on the title were removed from the keywords.
Acronyms were put in bold, even though it was not requested by the Journal, so as to ease the
reading process by highlighting the numerous acronyms found throughout the paper. This was
suggested by another reviewer and the authors found it to be a pertinent idea.
References, regarding the use of histology to assess the precision of determining a sexual
maturity phase through macroscopic criteria, were added.
The suggestion to delete the sentence (page 11) introducing Table 2 was applied since Table 2
had, indeed, already been introduced previously.
The description for the L50, previously in the discussion, was moved to the M&M section.
The suggestion to summarize the discussion was taken into account and a few areas were
shortened where possible.
“Figure 9 : Why was macroscopic maturity preferred to microscopic analysis?”
In this study, the aim was to underline the misclassifications when using the more classical
macroscopic method, and thus point out the importance of using histology coupled with
stereology.
As suggested by the reviewer, the ungutted but ovary-free weight was used to calculate the
GSI. Modifications in the M&M (formula), Figure 9 and values throughout the text were changed
to fit the new results.
Regarding whether the GSI estimated for “all sampled individuals” was for males or females,
and as stated in throughout the text, only females were sampled, thus the GSI found within this
document is for females only. However, the term ‘female’ was added in the result section (4.8)
so as to remind the readers of that fact.
All modifications may be found highlighted in the text.
Please see attachment

This manuscript is a resubmission of an earlier submission. The following is a list of the peer review reports and author responses from that submission.
Round 1
Reviewer 1 Report
Comments for Authors
This paper suggested a method based on quantitative histology (stereology) to estimate the sexual maturity phase of individuals, which allowed for a detailed description of the oogenesis cycle for the European Plaice, and got rich and high quality results.
My comments as follow:
Abstract: accepted
Introduction: accepted
Materials and methods: has scientific merits
Result: present clearly
Discussion: accepted
Conclusion: accepted
Question: Is this method applicable to other fish?
No comments.
Reviewer 2 Report
Dear Authors,
The reviewed article describes the possibility of creating a new system for determining the maturity of the gonads based on macroscopic and microscopic observations in the European Plaice.
The work has been constructed in a logical way, presenting the findings and evaluation criteria, and then presenting an alternative. The research material consisted of fish caught accidentally during commercial fishing. Each time, a dozen fish were collected, and twice as many during spawning. The time of collecting the material is 2017-2019. The results described in detail the microscopic appearance of the female gonads of the test species.
There are many words in the work, probably incorrectly translated from the national language of the authors, which slightly changed the meaning of the sentences during the translation. Please verify the work in the field of vocabulary and its correct use.
Detailed notes:
Latin name of species should be written with italic (ex in discussion).
WKMATCH WKASMSF – what does it mean? (description of table 1).
In Fig. 2a – you presented fish with an open abdominal cavity – very poor quality of presenting data. Please remove the background, upgrade the contrast and mark the gonads. At this moment reader can see the fish and something white – probably meat. There is no visible ovary. Please, correct it and add scale (and unify the scales style in the article).
The figure below – should have its letter. The presentation of these gonads is unacceptable. Do not attach photographs without scale – the coin is a great idea but in normal newspapers, not in scientific work. Correct it or remove this photo.
ALL TEXT: Please marked all abbreviations in text with different fonts or sizes of letters or italic or bold – it is hard to understand the sentences with this kind of abbreviation.
ALL FIGURE: I suggest moving the letters of each photo from the left upper corner to the left bottom corner and adding a white background. In the attached file all of them blend in with the background.
Why did you choose to write about all of the phases of oocytes based on the new abbreviation style presented in Table 2 and not based on Table 1 terminology? You have all the necessary parts – as a sub-phase, phase etc. Why does doubling work for the readers? It would be a lot easier to understand to keep only one table, and description of oocytes written as a full name and not this abbreviation which, by the way, you almost do not use in the results. Do not introduce more abbreviations because the work is no longer understandable.
Due to linguistic, visual and substantive inaccuracies, I suggest withdrawing the article in this form, making corrections, and then attaching it again (in this journal, corrections are allowed up to 3 days, which is not enough time).
Sincerely,
R.
Sentence constructions and vocabulary used in the work do not always correspond to the specialist vocabulary that should be used in scientific work.
Reviewer 3 Report
The manuscript is an attempt to provide information on the reproductive biology of the flatfish Pleuronectes platessa. The authors failed to correctly interpret the ovarian histology. Oocytes in fig. 4a, b, c and d are atretic oocytes as shown by the lack of the zona radiata and cytoplasm degeneration. Oocytes in fig. 4d and e are not at the migratory nucleus stage and the peripheral localization of the nucleus is due to the fact that the oocytes are degenerating. The migration of the nucleus at the animal pole occurs after the completion of vitellogenesis and it represents the beginning of the final oocyte maturation phase. The reported size at first maturity is meaningless in absence of the calculation of the maturity ogive showing the size at which 50% of the females are mature. In the present form the manuscript provides very limited information on the reproductive biology of the species and needs a carefull reconsideration of the histological interpretation.